# Influence of Lab Adapted Natural Diet and Microbiota on Life History and Metabolic Phenotype of *Drosophila melanogaster*

**DOI:** 10.3390/microorganisms8121972

**Published:** 2020-12-11

**Authors:** Andrei Bombin, Owen Cunneely, Kira Eickman, Sergei Bombin, Abigail Ruesy, Mengting Su, Abigail Myers, Rachael Cowan, Laura Reed

**Affiliations:** Department of Biological Sciences, University of Alabama, Tuscaloosa, AL 35487, USA; opcunneely@crimson.ua.edu (O.C.); kmeickman@crimson.ua.edu (K.E.); sbombin@crimson.ua.edu (S.B.); amruesy@crimson.ua.edu (A.R.); msu1@crimson.ua.edu (M.S.); amyers24@crimson.ua.edu (A.M.); racowan@crimson.ua.edu (R.C.)

**Keywords:** microbiota, 16S, metabolic phenotype, host microbiota interaction

## Abstract

Symbiotic microbiota can help its host to overcome nutritional challenges, which is consistent with a holobiont theory of evolution. Our project investigated the effects produced by the microbiota community, acquired from the environment and horizontal transfer, on metabolic traits related to obesity. The study applied a novel approach of raising *Drosophila melanogaster*, from ten wild-derived genetic lines on naturally fermented peaches, preserving genuine microbial conditions. Larvae raised on the natural and standard lab diets were significantly different in every tested phenotype. Frozen peach food provided nutritional conditions similar to the natural ones and preserved key microbial taxa necessary for survival and development. On the peach diet, the presence of parental microbiota increased the weight and development rate. Larvae raised on each tested diet formed microbial communities distinct from each other. The effect that individual microbial taxa produced on the host varied significantly with changing environmental and genetic conditions, occasionally to the degree of opposite correlations.

## 1. Introduction

The holobiont theory states that a host and its commensal microbiota possess a metagenome that expresses a synergistic phenotype, which is subjected to evolutionary forces as one complex organism [1]. A phenotype of this unit could be varied by genome modifications of the host, and its commensal bacteria. Metagenomic changes induced by bacteria have more potential for genetic variability and could arise by altering dominant species of bacteria, and acquisition of new strains of microorganisms from an environment [2].

Gut microbiota is one of the most important factors shaping metabolic phenotype and, as a consequence, is a key element in the development of metabolic and autoimmune diseases, cancer, and asthma [3,4]. Alterations in gut microbiota biodiversity and community structure are correlated with the development of the obese phenotype [5,6]. Transfer of the microbiota from an obese to a lean, axenic (microbiota free) individual significantly increases weight gain and adiposity, compared to axenic mice colonized with microbiota community from a lean organism [5,7,8]. These results suggest that obesity can be transferred from one individual to another; therefore, exhibiting some characteristics of an infectious disease [8].

Many species of *Drosophila* use rotting fruit as their primary host, including *Drosophila melanogaster*, and primarily consume the microorganisms growing in the rot. During colonization of a rotting fruit, *Drosophila* inoculates the substrate with their microbiota [9]. In addition, when females deposit embryos on a substrate, parental microbiota resides on the chorion of the egg [10]. The initial microbiota is gained by the first instar larvae through consumption of the chorion [10]. Later, during feeding, larvae acquire additional microbiota from the environment, representatives of which remain in the larval and pupal intestine until eclosion of the adult fly [11]. Wong et al. [12] showed that parental microbiota transferred with the chorion of the egg could modify the microbial community composition in a food substrate and in the offspring. In addition, the transfer of axenic *Drosophila* on food substrate would change the food microbial community to resemble the symbiotic microbiota composition that would develop in the host [12].

Symbiotic microbiota play an important role in *D. melanogaster* development and metabolic phenotype. Axenic flies have a longer development time, lower weight, protein, and glycogen content but higher free glucose and triglyceride levels [13,14,15]. There are similar negative phenotypic effects on axenic *D. melanogaster* and on those being raised on harmful diets such as high sugar and high fat [13,15,16,17,18]. Although axenic *Drosophila* consume less food, their energy storage indices (triglyceride, glucose, glycogen, and trehalose levels) stay significantly higher than that of conventional flies [19]. In addition, the presence of the commensal microbiota allows *Drosophila* to maximize their lifespan and reproductive output [4]. Dobson et al. [14] showed that the abundance of specific bacterial taxa is associated with a change in metabolic phenotype: *Acetobacter*, *Gluconobacter* and *Komagataeibacter* are negatively correlated with a fly’s energy storage index. *Lactobacillus* is positively associated with triglyceride concentration and *Achromobacter* and Xanthomonadaceae have a positive correlation with glycogen levels [20,21]. In addition, *Drosophila* microbiota varies across genetic backgrounds, which makes it possible to establish associations between a host’s genes and microbiota dependent metabolic responses and particular symbiotic species [14,21,22].

Symbiotic microbiota allow *Drosophila* to overcome the nutritional limitation of their diet. Shin et al. [23] showed that axenic larvae raised on a casamino acid diet experienced a 90% body size reduction and were not able to survive to form pupae. However, the presence of only one bacterial species, *Acetobacter pomorum*, could restore survival and the normal rate of larval development via induction of the hosts’ insulin-like growth factor signaling [23]. Leitão-Gonçalves et al. [4] demonstrated that axenic flies express a strong preference for yeast-rich food due to their demand for essential amino acids. *A. promorum* and several *Lactobacilli* species are able to suppress the yeast appetite in *Drosophila* and shift the flies’ nutritional preference toward high-sugar concentration diets [4]. Their change in nutritional preference may be explained by competition between the host and its symbiotic microbiota for available sugars and through production of essential amino acids by the microbial community [4].

Microbiota composition of lab dwelling *Drosophila* primarily consists of Acetobacteraceae, Enterococcaceae, and Lactobacillaceae families [22,24]. Within these families, the influence of *Acetobacter tropicalis*, *Enterococcus faecalis*, *Lactobacillus brevis*, and *Lactobacisul plantarum* on a host’s metabolic phenotype has been studied more than others [15,22,23]. However, the microbiota of wild fly populations may differ in the diversity and the abundance of dominant species [24]. In the lab, *Drosophila* raised on fruits still exhibit a more complex and diverse community of symbiotic microbes compared to conventionally raised flies [25]. Lab food preservatives, especially methylparaben sodium salt (moldex), largely contribute to the difference between natural and laboratory associated microbiota communities [26]. Therefore, studying the evolutionary relationship of *Drosophila* and its microbiota, and the symbiont’s influence on fly’s metabolic phenotype only on standard lab microbiota, may be insufficient to understand the natural relationship and coevolution of the fly and its microbiota.

With this work we wanted to address a series of specific questions and hypotheses:

**Hypothesis 1.** 
*How does a natural diet with a naturally occurring and/or maternally inherited community of microbes influence the life history and metabolic phenotypes of flies relative to a standard lab diet? Is there genetic variation in the phenotypic response to nutritional change and dietary and parental microbiota availability? We hypothesized that larvae raised on a natural diet will exhibit different life history traits and metabolic phenotypes, comparing with larvae raised on a standard lab diet. We hypothesized that the presence of a maternally transmitted microbiota will significantly impact larvae phenotypes, and that this impact may vary across dietary treatments. Given prior findings on the roles of genetic variation on metabolic phenotypes, we hypothesized that there is genetic variation in phenotype that interacts with the dietary conditions and the availability of maternally transmitted microbiota.*


**Hypothesis 2.** 
*Will the symbiotic microbiota community of the larvae raised on the natural diet be different from the lab food raised larvae? Will the presence of maternally inherited microbiota influence the formation of microbial communities? Is the microbiota community variable across host genotypes? We hypothesized that the gut microbial community composition and diversity will vary substantially across both dietary and parental microbiota conditions. We hypothesized that the maternally transmitted microbiota will have “founder effects” in the formation of the larval gut microbiome. We hypothesized that the composition of the microbial community will exhibit variation with host genotype.*


**Hypothesis 3.** 
*Will specific microbial taxa and/or microbiota communities as a whole influence the larvae phenotypes differently across diets and genotypes? We hypothesized that some microbial taxa will have consistent correlations with host phenotype across diets and treatments while others will have a diet or treatment specific relationship.*


## 2. Materials and Methods 

### 2.1. Food Preparation

Under hot and humid conditions on 28 August 2017, we put approximately 200 peaches outdoors at 33.2598614° latitude and −87.4919516° longitude (Tuscaloosa area, Alabama) and allowed them to decay for six days. Peaches were selected as fruits that are locally grown in Alabama and therefore might provide nutritional conditions experienced by local *Drosophila* groups. The six day period was chosen as our preliminary study indicated that this period of time produced a nutritional media that supported the development of more flies from first instar to adult stages than two, four, eight, or twelve-day periods. On 3 September 2017, the fruits were collected, manually ground, and stored in freezers at −20 °C. Peach food (PR) preparation protocol was the following: we allowed approximately one liter of the peach food to thaw, homogenized it with an immersion blender, and distributed it into vials, with approximately 10 mL of food per 47 mL “narrow” *Drosophila* vial. In order to prepare autoclaved peach food (PA), vials containing peach food were autoclaved for 25 min at 121 °C. Regular *Drosophila* lab food (R) was cooked according to the protocol described in previous works. Briefly, it was composed of cornmeal (60 g), yeast (12 g), agar (6 g), water (1 L), tegosept (13 mL), propionic acid (6 mL), and molasses (51 mL). [18,27].

To ensure that the autoclaved peach food did not contain any live microorganisms, we used 1 g of autoclaved and non-autoclaved peach materials and diluted them in 9 mL of sterile phosphate-buffered saline solution (PBS) [4,28]. Then, we performed a serial dilution in PBS to get food dilutions [29]. We mixed 1 mL of each dilution with standard methods agar (criterion) via the pour plate method [29]. The agar was prepared according to the manufacturer’s directions. Samples were incubated at 35 °C for 48 h and visually evaluated for the presence of microbial colonies [30]. The independent variable for the diet component will be referred to as D.

### 2.2. Drosophila Stocks and Husbandry

We used 10 naturally derived genetic lines created by the DGRP2 project: 142, 153, 440, 748, 787, 801, 802, 805, 861, and 882 [31,32]. These genotypes were chosen as they showed a large range of reaction norms of pupae weight across diets in the previous studies of the Reed’s lab and our preliminary study. We chose to focus only on *Wolbachia* positive stocks to avoid the confounding factors of phenotypic variation due to *Wolbachia* infection status, instead aiming to evaluate the influence of broad bacterial community on larvae phenotype. Stocks (on a molasses-based lab diet as described above) and experimental flies were maintained at constant temperature of 25 °C, 50% humidity, and 12-h light/dark cycle [16,18,27]. The independent variable for the genetic component will be referred to as G.

### 2.3. Drosophila Embryos Sterilization

In order to remove parental microbiota, we sterilized 12-h old embryos with subsequent two-minute washes in 2.5% active hypochlorite solution, 70% ethanol solution, and sterilized distilled water [4,28]. After sterilization, embryos were placed on the apple agar plates, and incubated until the first instar stage under fly-rearing conditions described above [33]. The non-sterilized control embryos (NS) were allowed to develop for 24 h (until the 1st instar larvae stage) on the apple agar plates, on which they had been deposited. In order to demonstrate that sterilized embryos did not possess parental microbiota, 20 sterilized 1st instar larvae were collected and grinded in 200 μL of the sterile PBS using a mechanical homogenizer. The resulting mixtures were plated on nutrient and standard method agars (criterion). The agars were prepared according to the manufacturer’s directions. Plates were incubated for 48 h at 35 °C and visually evaluated for the presence of microbial colonies [30]. NS larvae were used as the positive control according to the same procedure. All flies’ stocks were maintained in the lab for more than 100 generations on the cornmeal-molasses diet in a common incubator with controlled humidity and photoperiod. Therefore, any difference in parental microbiota composition observed between genetic lines at the initiation of the experiment are likely to be attributed to the genetic and not environmental factors. The independent variable for the sterilization treatment component will be referred to as T.

### 2.4. Larvae Rearing and Collection

In three separate time periods (30 days apart), 50 sterilized and non-sterilized larvae, of each genetic line were put in at least three vials of PA, PR, and R food, each. The independent variable for the time period (round) component will be referred to as Ro. Larvae were allowed to develop until the late third instar wandering stage (when they stopped moving but before the pupation started) and then collected in micro centrifuge tubes with sterile Ringer’s solution and immediately stored at −20 °C [33]. Each vial was checked for the presence of larvae, at the right developmental stage, four times per day at 9 am, 11 am, 2:30 pm, and 5 pm for 18 days after larval colonization. Prior to metabolic assays, larvae were inspected for the presence of any damage (damaged ones were removed), and cleaned with at least two washes in a sterile Ringer’s solution. Since we tried to minimize the experimental error potentially introduced by variation in time between individual collections of larvae and between genetic lines, larvae were frozen immediately after collections (larvae of the same diet/genetic/line/treatment/round were pulled together across the vials). We were not able to determine the sex of the frozen larvae and therefore did not account for it in our assays. Thus, there may have been sex-specific responses to these experimental conditions we were unable to account for in this study.

### 2.5. Measuring Experimental Phenotypes

Survival: The total number of third instar larvae collected per vial was divided by the number of first instar larvae used to seed the vial (50) to evaluate the proportion of larvae that survived till the late 3rd instar stage (an average of 11 samples per unique diet/treatment/genetic line combination). Developmental rate: The developmental rate, in days, was calculated for each larva individually, from the day it was put in the food vial to the collection date. We then calculated median developmental time per vial and used it in our statistical analysis (on the average nine samples per unique diet/treatment/genetic line combination) [11]. Weight: In order to measure the dry weight, larvae were taken from the −20 °C freezer, allowed to reach room temperature and placed in a VWR standard oven at 37 °C overnight. After drying, each larva was weighed individually using Mettler Toledo XS 105 microbalance. Weights were recorded with LabX direct software v. 2.2 (on the average 27 samples per unique diet/treatment/genetic line combination). Triglyceride: With the exception of four samples (due to low survival of larvae of certain diet/treatment/genetic line/round combinations), we homogenized 10 larvae per sample to determine total triglyceride concentration using the Sigma Triglyceride Determination Kit [18,34,35,36]. Results were adjusted to represent the average triglyceride concentration per milligram of dry larval weight (on the average six samples per unique diet/treatment/genetic line combination). Protein: Protein concentrations were quantified using the Bradford’s method with 10 homogenized larvae per sample (with the exception of 3% of the samples in which we used one to nine larvae, due to especially low survival rates of the specific groups) [18,37]. Protein concentrations were averaged to represent the protein concentration per milligram of dry larvae weight (on the average 12 samples per unique diet/treatment/genetic line combination). Glucose: For most of the samples, combined trehalose and glucose concentrations were quantified via homogenization of 10 larvae (with the exception of 5% of the samples in which we used four to nine larvae) with subsequent overnight incubation in 1 μg/mL trehalase solution and further application of the Sigma Glucose Determination Kit [16,18,38]. Glucose concentrations were averaged and adjusted to represent the amount of glucose per milligram of dry weight (on the average six samples per unique diet/treatment/genetic line combination).

To assess the triglyceride, protein and glucose concentrations in the diets, we used freshly unfrozen food and unfrozen food that was incubated in the food vials for seven days at 25 °C, in the same incubator as the experimental fly stocks, where larvae development was taking place. For the food assays, we used the same procedure as for the larvae but with 12.35 mg of the food sample. For the glucose assay, instead of incubating samples in trehalase, we incubated them in 100 μL of 1 mg/μL solution of invertase (to convert sucrose into glucose; Ward’s Natural Science) overnight at 37 °C. For analysis, the results were adjusted to represent the amount of the measured compound per milligram of the sample. These results can be viewed in Appendix A.

### 2.6. DNA Extraction and Sequencing

DNA was extracted from 10 larvae with the Qiagen blood and tissue DNA extraction kit according to the standard protocol, with overnight incubation of the samples in proteinase K at 56 °C. Prior to sequencing, the quality and purity of DNA extractions were confirmed with PCR, including negative controls, with the same primers used for the sequencing protocol and gel electrophoresis. DNA extractions were used for sequencing the V4 region of the 16S ribosomal RNA subunit (16S rRNA), which was performed in the Microbiome Core Facility of The University of Alabama in Birmingham, AL according to the previously published method on the Illumina MiSeq platform [39].

Trimmomatic-0.36 [40] was used to process demultiplexed DNA sequences. Standard Illumina-specific barcode sequences, all sequences with less than 36 bases, and leading and trailing low-quality bases were all removed using the default settings of the Trimmomatic-0.36 program. The USEARCH-fastq_mergepairs tool was used to combine forward and reverse readings. All reads with an expected error greater than 1 were removed, and chimeric reads and singletons. Combined readings without a merging pair were filtered using fastq_filter command. The -unoise3 tool was used to cluster readings into zero-radius OTU (ZOTU) 100% identity. OTUs were then designated with the lowest taxonomic rank using the UCULT algorithm implemented in QUIME 1.9.1 [41,42] along with SILVA reference database version 132 [43]. Using SILVA v. 132 database, PyNAST [44] with default options was used for sequence alignment. The phylogenetic trees of ZOTUs and OTUs were assembled using the default options of QUIIME 1.9.1 with the FastTree program [45]. Alpha and beta diversities were rarefied with QUIIME 1.91 -single_rarefaction.py using the—subsample_multinomial option in order to subsample the replacements. Rarefaction for all samples was performed to the depth of 4500 reads. This was the lowest possible number of reads between samples.

### 2.7. Statistical Analysis

Data transformation. Normality tests, data transformations, and statistical models were done with JMP Pro 14.0. Phenotype measurements were tested for normality with the Shapiro–Wilk test and an outlier box plot. Only larvae survival showed normal distribution. Therefore, all other phenotypic measurements data were transformed. We performed a cube root transformation on the data for development rate and glucose by weight, a square root transformation on data for weight and protein by weight, and a log transformation on data for triglyceride by weight concentrations. The bacterial abundance was log(x+1) transformed for all parametric analyses. 

Bacterial Diversity. Alpha and beta diversities were computed in QIIME v. 1.9.1. To estimate alpha diversity, we used Shannon, Simpson, and PD Whole Tree metrices. As all of the alpha diversity indices were not normally distributed, we performed a pairwise comparison of them with a Wilcoxon rank-sum test in R v. 3.5.1 with “matrixTests” package v. 0.17 [46]. Beta diversity was estimated with Bray–Curtis and weighted Unifrac distances. The similarity between each sample’s beta diversity distance was evaluated via hierarchical clustering, applying a ward method for distance calculation, and visualized with a constellation plot in JMP v. 14.0.

Statistical modeling: In order to evaluate the contribution of each variable and their interactive effect to each phenotypic development, we used standard least squares model with model effects to include diet (D), genotype (G), sterilization treatment (T), and their specific interactive effect: D × G (diet-by-genotype), D × T (diet-by-treatment), G × T (genotype-by-treatment), and D × G × T (diet-by-genotype-by-treatment). In order to verify that the built model fits the data, a lack of fit test was performed. If the time period of the experiment (Ro) and/or the variance between the colorimetric assay runs (triglyceride, protein, and glucose) (P) produced a significant effect, these variables were included in the model’s effects, unless their addition caused the model to fail the lack of fit test. Thus, the models for larvae survival, development time, and weight were the following:$$\begin{array}{c} y_{i}\;=\;\beta_{0}+\beta_{1}D_{1i}+\beta_{2}G_{2i}+\beta_{3}T_{3i}+\beta_{4}{\left(D*G\right)}_{4i}+\beta_{5}{\left(D*T\right)}_{5i}+\beta_{6}{\left(G*T\right)}_{6i}\\ +\;\beta_{7}{\left(D*G*T\right)}_{i7}+\;\varepsilon_{i} \end{array}$$

For triglyceride by weight:$$\begin{array}{c} y_{i}\;=\;\beta_{0}+\beta_{1}D_{1i}+\beta_{2}G_{2i}+\beta_{3}T_{3i}+\beta_{4}{\left(D*G\right)}_{4i}+\beta_{5}{\left(D*T\right)}_{5i}+\beta_{6}{\left(G*T\right)}_{6i}\\ +\;\beta_{7}{\left(D*G*T\right)}_{i7}+\;\beta_{8}\;P_{8i}+\varepsilon_{i} \end{array}$$

Additionally, for protein and glucose by weight:$$\begin{array}{c} y_{i}\;=\;\beta_{0}+\beta_{1}D_{1i}+\beta_{2}G_{2i}+\beta_{3}T_{3i}+\beta_{4}{\left(D*G\right)}_{4i}+\beta_{5}{\left(D*T\right)}_{5i}+\beta_{6}{\left(G*T\right)}_{6i}\\ +\;\beta_{7}{\left(D*G*T\right)}_{i7}+\;\beta_{8}\;P_{8i}+\beta_{9}R_{9i}+\varepsilon_{i} \end{array}$$
where $y_{i}$ is the response, β values are constants, and $\varepsilon_{i}$ is a random error term.

All models for interactive effects of diet, genotype, and treatment were done with all 10 genetic lines, with the exception of glucose, which was done without 861 due to the low survival rate of this genetic line.

In order to verify that the built model fits the data, a lack of fit test was performed. Only the models with non-significant lack of fit *p*-value were kept. To assess the pairwise difference between diets (R vs. PR and PR vs. PA) and treatments (NS vs. S), we used the least square model with one main explanatory variable of interest (diet or treatment). We also included time period and assay plate variance as additional explanatory variables if they produced a significant effect. The pairwise comparisons were performed with a post-hoc Student’s *t*-test.

Bacterial abundance: In order to evaluate if the diet and treatment could serve as categorical predictors for classification of the larvae bacterial samples, we performed discriminant analysis at phylum, class, order, family, and genus taxonomic levels, and at the level of individual ZOTUs. The results were visualized with a canonical plot in JMP v. 14.0 (JMP manual). For the 10 most abundant representatives of each taxonomic level, we applied the linear covariance method for the discriminant analysis, which allowed us to visualize the covariates in the form of vector rays. Using this method allowed us to represent which of them drove the separation of the clusters (JMP manual). The length of the ray is correlated with the strength of the impact that it produced on the samples to be separated in the vector direction, on a canonical plot. When we ran the analysis with all identified taxa, we applied a wide linear method for the discriminant analysis. To compare the abundance of bacterial taxa between the diets and treatments, we performed a Wilcoxon test as described above. The *p* values were adjusted for the false discovery rate with Benjamini–Hochberg correction and added to data tables as FDR p (False discovery rate adjusted *p* value). The threshold for the value of FDR p that should be considered significant could be subjective and vary from 0.25 to 0.05 among microbiology studies [47,48]. To evaluate the interactive effect of the variables on the abundance of each identified bacterial taxa, we used the three-way linear interaction model.
$$\begin{array}{c} y_{i}\;=\;\beta_{0}+\beta_{1}D_{1i}+\beta_{2}G_{2i}+\beta_{3}T_{3i}+\beta_{4}{\left(D*G\right)}_{4i}+\beta_{5}{\left(D*T\right)}_{5i}+\beta_{6}{\left(G*T\right)}_{6i}\\ +\;\beta_{7}{\left(D*G*T\right)}_{i7}+\beta_{8}R_{8i}+\varepsilon_{i} \end{array}$$

In order to identify the correlations between larvae phenotypes and bacterial abundances, we found the mean phenotype for each combination of diet, genetic line, treatment, and round. Spearman’s rank correlation between the microbial abundances and tested phenotypes was calculated with Hmisc v. 4.3-0 in R v. 3.5.1, with the adjustment of *p* values for FDR p as described above. We also tested the possible interactive effect of each identified bacterial taxa and one of the independent variables on the formation of the tested phenotypes according to the formula:$$y_{i}\;=\;\beta_{0}+\beta_{1}x_{1i}+\beta_{2}x_{2i}+\beta_{3}{\left(x_{1i}*x_{2i}\right)}_{3i}+\;\beta_{4}R_{4i}+\varepsilon_{i}$$
where, $x_{1}$ was the abundance of the microbial taxa, $x_{2}$ was one of the independent variables (D, G, or T), and R was the time component. Development, weight, triglyceride, protein, and glucose were normalized with log, square root, log, log, and cube root, transformations respectively.

## 3. Results

### 3.1. There Are Substantial Differences in the Life History and Metabolic Phenotypes for Larvae Raised on a Natural Peach Diet with a Naturally Occurring and/or Maternally Inherited Community of Bacteria Relative to a Standard Lab Diet

#### 3.1.1. Larvae Raised on a Natural Diet Exhibited Different Life History Traits and Metabolic Phenotypes from Larvae Raised on a Standard Lab Diet

Survival: The proportion of larvae that survived on the lab diet was significantly higher than the larval survival on a natural peach diet regardless of sterilization (Figure 1A, Table 1 and Appendix A). The undisturbed bacterial community of the peach diet produced a significant positive effect on larvae survival, when compared with the autoclaved peach diet (Figure 1A, Table 1 and Appendix A). Development rate: Within both controlled and sterilized treatments, larvae developed faster on the regular lab diet compared to the natural diet and faster on the regular peach food compared to the autoclaved peach food (Figure 1B, Table 1 and Appendix A). Weight: The third instar larvae raised on the lab food were significantly heavier than those that were raised on the natural diet (Figure 1C, Table 1 and Appendix A). Among the peach diets, larvae consuming the autoclaved diet were significantly lighter (Figure 1C, Table 1 and Appendix A), especially for larvae deprived of parental microbiota, suggesting that under natural nutritional conditions, microorganisms may facilitate growth and weight gain of the larvae. Triglyceride: Although fresh and incubated R food had higher triglyceride concentrations than the PR food (Appendix A), larvae raised on the PR diet had significantly higher triglyceride concentrations by weight than those that were raised on a lab diet, independent of sterilization treatment (Figure 1D, Table 1 and Appendix A). Incubated autoclaved peach food had significantly higher triglyceride content compared with regular peach food (Appendix A) and produced larvae with higher triglyceride concentration by weight (Figure 1D, Table 1 and Appendix A). Protein: Independent of the treatment, larvae raised on the regular lab food had higher protein by weight concentrations compared to larvae raised on peach food (Figure 1E, Table 1 and Appendix A). Larvae raised on the PA diet had significantly higher protein by weight levels compared to PR raised larvae, but only in the absence of the parental bacterial community (Figure 1E, Table 1 and Appendix A). Evaluating the protein quantity in fresh food itself, we found significantly higher protein concentration in the regular lab diet compared to the peach diet (Appendix A). Between fresh peach diets there was no significant difference in protein (Appendix A). However, after incubation, autoclaved peach food had significantly less protein than regular peach food (Appendix A). Glucose: Larvae raised on a lab food diet had significantly higher glucose concentrations than larvae raised on the peach food diet (Figure 1F, Table 1 and Appendix A). Larvae that consumed PR food had lower glucose by weight concentrations compared with larvae raised on the autoclaved version, which was consistent with our hypothesis (Figure 1F, Table 1 and Appendix A). Interestingly, the fresh peach food itself had a significantly higher glucose concentration than the lab food (Appendix A). However, after incubating the peach food, the concentration of glucose was lower in PR than in both R and PA diets (Appendix A), suggesting a strong impact from the live bacterial community.

#### 3.1.2. We Observed That the Presence of a Maternally Transmitted Bacteria Significantly Impacted Larvae Phenotypes, and That Impact Varied across Dietary Treatments

Survival: The presence of the parental bacteria significantly enhanced overall survival of larvae on the autoclaved peach diet but did not produce a significant effect on larval survival on the lab diet or the non-autoclaved peach diet (Figure 1A, Table 2 and Appendix A). This suggested that the availability of bacterial taxa was necessary for a successful transition of the larvae through the instar stages under natural nutritional conditions, and that these bacterial taxa could be acquired from the food substrate if available and/or inherited maternally. Development rate: Presence of parental bacteria on the peach diet reduced the number of days necessary for larvae to reach the third instar stage on the autoclaved peach diet but not on the regular lab diet (Figure 1B, Table 2 and Appendix A), suggesting that under natural nutritional conditions, maternal microbes might influence the larval developmental rate independent of bacteria acquired from the food substrate. Weight: Maternally inherited bacteria produced a significant positive effect on larval weight on all of the tested diets (Figure 1C, Table 2 and Appendix A). This indicated the universality of their influence on larval growth across food substrates. Triglyceride: Parental bacteria did not influence the triglyceride concentrations significantly on any diet (Figure 1D, Table 2 and Appendix A). Protein. Evaluating the role of parental bacteria, we observed that sterilized larvae had higher protein by weight concentrations but only on the PA diet (Figure 1E, Table 2 and Appendix A). This suggested that the core bacteria involved in a natural metabolic phenotype formation might be inherited or acquired from the environment. Glucose: The parental bacteria reduced the glucose by weight concentrations only on PA food (Figure 1F, Table 2, and Appendix A), indicating that both parental and environmental microbial taxa might be sufficient to reduce glucose concentrations in larvae.

#### 3.1.3. Evaluating the Contribution of Tested Independent Variables on Larvae Phenotypes, We Observed a Genetic Variation in Most of the Tested Life History Traits and Phenotypes that Interacted with the Dietary Conditions and the Availability of Maternally Transmitted Microbiota

Survival: All of the independent variables included in the model produced a significant effect on larval survival until the late third instar stage. Of the tested variables, diet was the strongest predictor of survival, followed by the interactive effect of the diet by treatment and genetic line (Table 3 and Appendix A). Development: The development rate of the larvae was significantly influenced by diet, genotype, and treatment (Table 3 and Appendix A). Among the specific interaction of these variables, only D × T and G × T produced a significant effect on development (Table 3 and Appendix A). Diet was the key factor that influenced the time necessary for the larvae to reach the late third instar stage and explained almost half of all variance followed by the genetic line (Table 3 and Appendix A). The combination of the rest of the variables was responsible only for 8.42% of variation in developmental time (Table 3 and Appendix A). 

Weight: All of the independent variables, with the exception of D × T, produced a significant effect on dry larval weight with diet being the best predictor, followed by genotype, and D × G interaction (Table 3 and Appendix A). Triglyceride. Once again, diet explained the largest portion of variance across all independent variables. The interactive effect of D × G was a better predictor of triglyceride concentrations than the genotype (Table 3 and Appendix A). Protein: In contrast with other measured phenotypes, the variance explained by the model was predominantly evenly distributed across the independent variables, with genotype having the highest predicting power followed by the D × G interactive effect (Table 3 and Appendix A). Glucose. Diet was the strongest predictor of larvae glucose concentrations (Table 3 and Appendix A). Other variables that produced a significant effect on glucose concentrations were treatment, D × G, and D × T (Table 3 and Appendix A).

### 3.2. The Symbiotic Bacterial Community Composition of the Larvae Raised on the Natural Diet Was Different from the Lab Food Raised Larvae and Was Influenced by Maternally Inherited Bacteria and the Host’s Genotype

#### 3.2.1. The Gut Bacterial Community Composition and Diversity Varied Substantially across Dietary and Treatment Conditions

Alpha diversity: We characterized a total of 6763 unique ZOTUs across the whole dataset with the number of ZOTUs per sample ranging from 55 to 886. The total number of reads per sample ranged from 4685 to 908,308. Of the ZOTUs that could be assigned a taxonomic classification, we identified 134 classes, 218 families, and 394 genera represented across the samples. The response of alpha diversity to changing diets varied with the larval sterilization treatment. For NS larvae, we found that the Shannon index of larvae raised on PR and R diets was significantly higher than those raised on the PA diet (Appendix A). All other comparisons were not significant. However, if the embryos were subjected to sterilization, the microbial species richness of larvae raised on the PA diet was significantly higher than larvae raised on the PR diet (Appendix A). In addition, there was no significant difference in bacterial species richness between larvae raised on regular or peach regular diets (Appendix A). We observed the exact same pattern for the PD whole tree index. Larvae raised on any diet were not significantly different in Shannon’s index (Appendix A).

Beta diversity: The hierarchical clustering of the Bray–Curtis distances indicated that the most distant bacterial communities were formed between larvae raised on the R and PR diet (Figure 2A). This pattern held true for both sterilized and non-sterilized larvae (Figure 2A). Clustering weighted Unifrac distances suggested that PR and R diets might produce symbiotic bacterial communities that were phylogenetically distant from each other, especially if the parental microbiota had been removed (Figure 2B).

Taxa composition: Applying discriminant analysis on the ten most abundant bacteria at each taxonomic level revealed which organisms were largely responsible for the differentiation of the bacterial composition on the canonical plot, based on diet. Thus, PR food is largely defined by the abundance of Cyanobacteria at the phylum level (Appendix A), Epsilonproteobacteria at the class level (Appendix A), Streptophyta at the order level (Appendix A), Leuconostocaceae sequences at the family level (Appendix A), and *Leuconostoc* at the genera level (Figure 3A, Appendix A). In turn, the lab diet was defined by Firmicutes, Bacilli, Lactobacillales, Lactobacillaceae, and *Lactobacillus* respectively (Figure 3A, Appendix A). Interestingly, when we considered only the 10 most abundant organisms at each taxonomic level, we did not see a full separation between R and PA diets unless the larvae were sterilized (Figure 3A, Appendix A). If parental microbiota were removed, the differentiation of the PA diet was led by Actinobacteria at the phylum level, Actinobacteria and Alphaproteobacteria at the class level, Clostridiales and Rickettsiales at the order level, Lachnospiraceae and Rickettsiaceae at the family level, and *Bacteroides* and *Wolbachia* at the genus level (Figure 3A, Appendix A). Including all identified bacterial groups in the discriminant analysis revealed that diet was a good predictor of bacterial taxa composition at the phylum, class, order, family, genus, and even individual ZOTU levels (Figure 3B, Appendix A).

Overall, for non-sterilized larvae the abundance of eight phyla, 12 classes, 20 orders, 27 families, 40 genera, and 141 ZOTUs were significantly different between PR and R food (Appendix A). Comparing PR and PA food, we found that the abundance of 4 phyla, 6 classes, 9 orders, 16 families, 20 genera, and 76 ZOTUs were significantly different (Appendix A). Lastly, we observed the significant difference for the abundance of four phyla, three classes, five orders and families, three genera, and 27 ZOTUs between R and PA food (Appendix A). This indicated the minimal difference between microbial communities of these diets to be consistent with the discriminant analysis.

In larvae lacking parental microbiota, we observed that the abundance of 6 phyla, 10 classes, 18 orders, 34 families, 37 genera, and 114 ZOTUs were significantly different between lab and peach diets (Appendix A). Comparing PR and PA diets, we found a significant difference in the abundance of seven phyla, 14 classes, 32 orders, 48 families, 67 genera, and 200 ZOTUs (Appendix A). R and PA diets were significantly different in the abundance of 5 phyla, 6 classes, 9 orders, 13 families, 18 genera, and 87 ZOTUs (Appendix A).

#### 3.2.2. The Maternally Transmitted Microbiota Influenced the Composition of the Larvae’s Symbiotic Bacterial Communities

Alpha diversity: S larvae had higher values for species richness, Shannon, and PD whole tree indexes on the PA diet (Appendix A). NS larvae had a significantly higher Simpson index value on the PR diet (Appendix A). All other comparisons were not significantly different.

Beta diversity: When comparing the difference between beta diversity metrics in NS and S treatments for each diet, we observed a distinctive clustering, based on the treatment of samples that were raised on PA food for Bray–Curtis distance (Figure 4A) and weighted Unifrac distance (Figure 4D). For the samples that were raised on R food, we observed the clustering for weighted Unifrac distance only (Figure 4F).

Taxa composition: The discriminant analysis indicated that the status of inheritance of the parental bacteria could serve as a good predictor for differentiation of the bacterial community as indicated with the canonical plot on all taxonomic levels (phylum, order, class, family, genus, and ZOTU; Figure 5, Appendix A). Among the 10 most abundant phyla that defined the differentiation of the NS community were Firmicutes and Bacterioidetes on the PA diet (Appendix A), Firmicutes on the PR diet (Appendix A), and Proteobacteria, Tenericutes, and Flusobacteria on the R diet (Appendix A). Phyla that were influential for differentiation of the S community were Actinobacteria, Fusobacteria, and Cyanobacteria on the PA diet (Appendix A), Actinobacteria, Tenericutes, and Flusobacteria on the PR food (Appendix A), and Planctomycets and Bacterioides on the R diet (Appendix A). Considering bacterial classes, the NS treatment was strongly defined by Bacilli on the PA (Appendix A), and the PR diets (Appendix A), and Alphaproteobacteria and Bacteroida on the R food diet (Appendix A). The sterilization treatment was mostly separated due to Actinobacteria on the PA diet, Alpha and Beta proteobacteria on the PR diet, and Gammaproteobacteria on the R diet (Appendix A). On R food, microbial communities from the sterilized and non-sterilized larvae were not fully separated on the canonical plot Appendix A). At the order level, the NS community was defined by Lactobacillales on the PA and the PR diets, and Actinomycetales and Rhodospirillales on the R food (Appendix A). Sterilized larvae were associated with abundances of Streptophyta on the PA food, and Burkholderiales and Rhodospirillales on the PR diet (Appendix A). On the R diet, 4 out of 10 tested orders were strongly associated with the S treatment (Appendix A). At the family level, the NS larvae were correlated with Lactobacillaceae on the PA and the PR diets and Acetobacteraceae on the R food diet (Appendix A). Sterilized larvae were defined by the abundance of Nocardiaceae on the PA food and Leuconostocaceae and Caulobactereceae on the R diet (Appendix A). At the genera level, NS was primarily separated by *Lactobacillus* on the PA diet and *Acetobacter* and *Agrobacterium* on the R food (Figure 5). On the PR food diet, 95% confidence ellipses almost overlapped, indicating that sterilization status might not be the decisive predictor for the abundance of the 10 most common genera (Figure 5). The S treatment was primarily defined by the abundance of *Leuconostoc* and *Gluconobacter* on the PA diet and *Lactobacillus* and *Leuconostoc* on the R diet (Figure 5).

Multiple microbial groups were significantly different in their distribution between NS and S treatments across the diets on all taxonomic levels. On regular food, the abundance of 4 phyla, 4 classes, 5 orders, 6 families, 9 genera, and 41 ZOTUs were significantly different (Appendix A). On the PR diet we saw a significant difference in the abundance of 1 phylum, 2 classes, 6 orders, 9 families, 11 genera, and 61 ZOTUs (Appendix A). The highest number of significantly different taxa was observed on the PA diet with 7 phyla, 12 classes, 21 orders, 34 families, 43 genera, and 148 ZOTUs (Appendix A).

#### 3.2.3. The Composition of the Microbial Community Exhibited Variation with Host Genotype, Which Further Exhibited a Significant Interactive Effect with Diet and Treatment

We also tested the influence of genotype and other variables’ interactive effect on the abundance of bacteria. At the phyla level, 14 were significantly influenced by genotype, three by D × G interaction, five by G × T, five by D × T, and six by D × G × T (Appendix A). Abundances of 30 classes were significantly influenced by genotype, eight by D × G, G × T, and D × T, and 10 by D × G × T interaction (Appendix A). Among the orders, an abundance of 46 was significantly influenced by genotype, 13 by D × G, 15 by G × T, 16 by D × T, and 15 by D × G × T (Appendix A). The abundance of 72 families was significantly influenced by genotype, 20 by D × G, 15 by G × T, 18 by D × T, and 23 by D × G × T (Appendix A). Lastly, genotype significantly influenced 94 genera, D × G influenced 44, G × T influenced 46, D × T influenced 30, and D × G × T influenced 45 genera (Appendix A).

### 3.3. We Identified Microbial Taxa That Exhibited Correlations with Host Phenotype across Diets and Treatments, with Many that Had a Diet, Treatment, or Genotype Specific Relationship

Across the NS larvae, we found four significant interactions on the R food, 14 on PR diet, and 14 on PA diet at the phylum level (Appendix A). For the S larvae at the same taxonomic level, we found eight significant interactions on the R diet, five on the PR, and two on PA food (Appendix A). At the class taxonomic level, 9 significant interactions were found on R food, 35 on PR, and 24 on PA diet (Appendix A). Considering S larvae, there were 23 significant correlations on the R diet, 11 on PR, and 4 on the PA diet (Appendix A). For NS larvae at the order level, we found 23 significant correlations on the R diet, 57 on PR, and 46 on PA diet (Appendix A). For S larvae we found 29 significant correlations on R food, 22 on PR, and 9 on PA diet (Appendix A). At the family level, we found 34 significant correlations on R food, 88 on PR, and 67 on PA diet for NS larvae (Appendix A). For S larvae, we observed 51 significant interactions on R food, 29 on PR, and 14 on PA diets (Appendix A). Across the genera, we found 40 significant interactions on R, 105 on PR, and 64 on PA diets, for NS larvae (Figure 6 and Figure 7, Appendix A). Considering S larvae, we found 76 significant interactions between tested taxa and phenotypes on R, 46 on PR, and 33 on PA diets (Figure 6 and Figure 7, Appendix A). At the level of individual ZOTUs, for NS larvae, we found 226 significant interactions on R, 283 on PR, and 225 on PA diets (Appendix A). For S larvae the number of significant interactions between ZOTUs abundances and larvae phenotypes were as follow 313 on R, 164 on PR, and 230 on PA diets (Appendix A).

We evaluated the interactive effect of the abundance of microbial taxa and diet, genotype, and treatment on forming the tested phenotypes. D × A produced a significant effect in 11 cases at the phylum level, in 26 at the class level, in 52 at the order level, in 72 at the family level, and in 95 cases at the genus level (Appendix A). We found a significant G × A interaction in eight cases at the phylum level, in 10 at the class level, in 39 at the order level, in 57 at the family level, and in 105 cases at the genus level (Appendix A). T × A produced a significant effect in 13 cases at the phylum level, in 27 at the class level, in 36 at the order level, in 60 at the family level, and in 87 cases at the genus level (Appendix A).

## 4. Discussion

### 4.1. Frozen Peach Food Was Capable of Providing Nutritional Conditions Similar to the Natural Ones and Can Preserve Key Microbial Taxa Necessary for Survival and Development of Drosophila Larvae

The reduction in survival, increase in development time, and increase in triglyceride concentrations, and decreased weight and protein concentrations of the larvae raised on the natural food compared with the larvae raised on the R lab food resembles the phenotype generated by a reduced protein diet. These findings correlate with our evaluation of the protein concentrations in different diets [49,50,51,52]. In addition, the adaptation of *Drosophila* to the lab environment was connected to increased weight and reduced stress tolerance [53,54,55]. Therefore, nutritional and pathogenic stresses associated with the natural food conditions could further contribute to the decrease in survival and development rate of larvae raised on the PR food compared to the standard diet [50,52,56,57].

The pattern regarding glucose concentration was more interesting. Freshly unfrozen peach food had a higher glucose concentration than regular lab food, but larvae raised on the PR diet had the lowest concentration compared to larvae raised on any other diet. This pattern was likely caused by the activity of naturally acquired microbes since it was shown that the presence of several microbial taxa that naturally associate with *Drosophila*, such as *Acetobacter*, is correlated with decreased sugars in fly food and *Drosophila* itself [14,15]. In addition, incubation of the PR food, even without the larvae, led to a drastic reduction of glucose concentrations compared to the R and PA diets.

It is well known that heat application can result in the decomposition of sugars and other macronutrients [58,59]. However, the difference between all phenotypes (with the exception of glucose) increased if the peach food diet was autoclaved and even more (with the exception of triglyceride) if the parental microbiota were not transferred to the autoclaved diet. This suggested that symbiotic bacteria played an important role in shaping the phenotypic change between PR and PA raised larvae. These findings are consistent with previous studies that showed that the presence of naturally associated bacteria was advantageous for *Drosophila melanogaster* and *Drosophila suzukii* on fresh fruit diets, which are also poor in protein content [50,57]. In fact, larvae raised on a PA diet closely resembled the phenotype of axenic larvae and axenic larvae raised under low protein nutritional conditions. Examples of this similar phenotype include lower survival and body size/weight [12,14,23], longer development time [13,23], and elevated glucose [15] and triglyceride concentrations [13,14]. Although, larvae raised on the PA diets exhibited significantly longer development time than on any other diet, we did observe minimal changes in the PA food nutrient concentrations during the incubation period (Appendix A). Therefore, it is unlikely that larval metabolic phenotype would be influenced by depletion of nutrients during these extra days.

### 4.2. Maternally Deposited Microbes Produced Positive Effects on Larvae that Were Raised on the Peach Diets

Interestingly, the presence of parental microbiota did not produce a significant effect on any of the tested phenotypes, when larvae were raised on the lab diet. Contrarily, on the peach diet, the presence of parental microbiota increased the weight and development rate even if the original peach microbiota were still present. These findings are consistent with the reports of beneficial effects, of the maternally deposited microbiota, for larvae on a fruit diet. These results also indicate the importance of considering an organism’s natural environmental conditions when addressing the questions about symbiotic relationships and evolutionary patterns [50,57].

### 4.3. Genotype Was One of the Key Factors that Influenced Larvae Phenotypes

It is important to note that although the described patterns were observed for the total experimental population of larvae, the genetic component still played a significant role in generating all but the glucose phenotype. In addition, consistent with previous research, we observed that D × G interaction played a significant role in forming the metabolic phenotype and contributed to the survival of the organism [16,36]. Furthermore, most of the tested phenotypes were significantly correlated with G × T and even D × G × T, indicating the importance of considering multiple factors to understand the development of complex traits.

### 4.4. Bacteria of the Larvae Raised on PR Food Exhibit a Distinct Community Structure

Multiple studies were performed to evaluate the gut microbiota composition of lab and wild populations of *Drosophila* [24,57,60,61,62]. Although most of them consistently report the prevalence of different members of Alphaproteobacteria, Bacilli, or Gammaproteobacteria in lab and wild populations, the relative abundance of the taxa, especially at lower taxonomic levels, often varies between studies [61]. In our work, larvae raised on the PR food diet formed a distinct community clearly separated from the larvae raised on the R food diet, as displayed on the canonical plot. We observed a higher prevalence of *Gluconobacter* and *Leuconostoc* and lower abundance of *Lactobacillus* in larvae raised on the PR diet compared with the R food diet [24,56,57,63], which is consistent with previous findings performed on natural populations of *Drosophila*.

However, it is difficult to judge how well the microbial community of our experimental larvae represents the microbial community of wild population, as a variety of factors could influence gut microbiota composition in flies, which certainly complicates comparisons between studies [56,61,64]. As such, it was shown that the gut microbial composition of lab reared flies might vary with diet (and even among the standard diets with the major carbohydrate source), genetic line, development stage temperature, and etc. [61,64,65,66]. The wild populations of gut microbiota in *Drosophila* were shown to vary with collection location and diet [56,60,62,67]. In other insects and wild populations of vertebrates, gut microbiota was shown to change even with seasonality [68,69,70,71].

In addition, the relationship between *Drosophila* gut microbiota during the developmental and adult stages is a subject of controversy between a few studies that compared those relationships [25,64,65]. Furthermore, to the best of our knowledge, the gut bacteria of the larvae from the natural populations were not assessed at all. This is likely due to the complexity of identifying *Drosophila* species during the larval stage. Therefore, we hope to provide the methodology for the possibility of exploring the effects of a natural diet, and the microbial community associated with it, in a controlled lab environment. This setting provides the opportunity to work not only with adult flies but also with larvae.

### 4.5. Community Structure of Symbiotic Bacteria Were Correlated with Diet, Treatment, Host Genotype, and Their Specific Interactive Effects

The development of symbiotic microbiota populations was shown to be correlated with the available nutrients present in the diet, the host’s genotype, and parental microbiota left on the chorion of the egg [12,61,64]. Complementary to the results reported by Jehrke [64], we also observed that the genotype of the host may influence the abundance of bacterial taxa more than the diet. Wong [12] reported that the bacterial population deposited on the *Drosophila* embryo may shift the symbiotic microbiota population of the offspring, even in the presence of bacteria that previously colonized the food substrate. We observed similar results in most cases.

However, among the 10 most abundant genera on the PR diet, the full separation of the S and NS larvae microbial community compositions was not present on the canonical plot indicating the possibility of a difference in the response of the lab and the natural microbial population to the presence of *Drosophila* parental bacteria. This differentiation was not likely caused by the nutrition composition of the food since the PA separation, represented on the canonical plot, between S and NS treatments was obvious in all cases. In addition, for beta diversity distances, the abundance of individual microbial taxa, and the correlations between the abundances of microbial taxa within the microbial community, represented patterns found in PA food that resembled the ones in the R food raised larvae, if parental microbiota were not removed (Figure 2 and Figure 3, Appendix A). Overall, consistently with previous studies, our findings indicated the dependency of relative bacterial abundances on all of the tested variables and demonstrated the interactive effect between these variables [12,61,64].

### 4.6. The Correlations between Microbial Taxa as Well as the Correlation between the Whole Microbial Community and the Host May Vary with the Diet and Other Environmental and Genetic Conditions

Genotype and gut microbiota composition are among the major factors that control the development of obesity traits [72]. Changes in some key microbiota populations are associated with the rapid expansion in the prevalence of metabolic syndrome [73]. Alterations in the gut microbiota community can modulate insulin secretion and sensitivity, thus contributing to diabetes susceptibility [74]. Moreover, previous research indicates that genetic variation considerably influences the gut microbiota composition [64,73,74]. However, most of the studies mentioned above have used less than ten genotypes to study the correlation between gut microbiota and the pathogenesis of obesity in mice. The challenges of using a mouse model involve relatively high expenses for husbandry and logistics [75,76,77].

*Drosophila melanogaster* is an exceptional model to study the effect of genotype on the phenotype formation, due to the variety of established tools such as *Drosophila* Genetic Reference Panel and The *Drosophila* Synthetic Population Resource. These resources offer a variety of diverse genotypes, with sequenced parental genomes, that allow for testing the microbiota effects across various genetic backgrounds and provides potential for studying genetic interaction between the host and its symbionts, and even mapping the specific genetic loci responsible for the interactions [21,31,78]. The phenotypic response to a diet modification often varies with the genotype [16,36]. In fact, diet-by-genotype (D × G) interaction may explain more variance than diet alone in the metabolic response of such traits, such as triglyceride and carbohydrate concentrations [36]. In addition, recent findings showed that genotype-by-diet interactions significantly influences metabolomic profiles; hence, laying the foundation for explaining the mechanism through which D × G influences metabolic traits [16,79]. Our results are consistent with previous research, in that phenotypic response varied significantly between genetic lines [16,79]. Genotype had a significant effect on survival, development rate, and triglyceride concentrations, and was the second-best predictor of weight and the best predictor of protein concentrations.

Obesity and type two diabetes are associated with elevated weight, high blood glucose concentrations, and excess accumulation of adipose tissue [80,81]. Consistent with recent studies linking *Lactobacillus* and *Coprococcus* to obesity in humans, our results show that these genera are positively associated with glucose concentrations [82,83,84,85]. In addition, previous research has shown an overall decrease in the abundance of *Firmicutes* in obese humans [86]. Similarly, we observed that the total abundance of *Firmicutes* is negatively associated with triglyceride concentrations. It should be noted that the correlation between metabolic phenotype and particular microbial taxa could vary between studies [5,87,88].

Consistently with previous *Drosophila* research, we observed that the abundance of microbial taxa was correlated with measured phenotypes. Acetobacteraceae was negatively correlated with larval glucose concentrations [20,21,61]. Additionally, *Acetobacter* was positively correlated with development time while *Lactobacillus* and *Firmicutes* were negatively correlated with it [20,21,89]. Previous work showed that *Acetobacter* species reduced triglyceride concentrations while most *Lactobacillus* species had no effect [13,21]. In contrast, our data shows that *Acetobacter* was not significantly correlated with triglyceride concentrations, and *Lactobacillus* showed a negative correlation. Consistent with Newell and Douglas [13], we found that *L. brevis* and *L. plantarum* were not significantly correlated with protein concentrations, but in addition, our results indicated that *Acetobacter* abundance was negatively correlated with protein concentrations.

Consistent with Jehrke [64], we observed that most of the correlations between the tested phenotype and abundance of bacteria were relatively weak. Weaker correlations observed with large sample sizes in microbiome research, while significant, failed to hold up to the use of stricter FDR values or other conservative p adjustment methods [47,64]. Expanding the analysis of the bacterial species abundance for each phenotype beyond the most dominant species, while providing a more complete overview of the correlation between tested phenotypes and microbial abundance, also raises FDR values as a result of increasing sample sizes [47]. Previous microbiome studies have dealt with high FDR values by accepting higher thresholds, so as to not miss possible correlations [47]. Since the level of FDR that should be tolerated is poorly defined and often widely variable compared to accepted *p*-values, its value is often seen as arguable [90]. Considering the large sample sizes used in our analysis, using a low FDR value may obscure important correlations between the tested phenotypes and their abundance of microbiota [47].

Some of the inconsistencies between our work and previous studies on the correlation of the abundance of microbial taxa and measured metabolic phenotype perhaps may be addressed to the interactive effect between the variables included in the experiments. Several studies showed that the contribution of the symbiotic microbiota to the host might be observed only in a diet dependent manner. For example, Shin et al. [23] showed that axenic *Drosophila* larvae would not be able to develop on a protein poor diet without activation of the insulin signaling pathway by its symbiotic microbes. Wong et al. [19] found diet-dependent differences in microbiota produced effects, including reduction of vitamin requirements on a low-yeast diet and suppression of lipid and carbohydrate storage on a high-sugar diet. Bing et al. [50] found that symbiotic microbiota of *D. suzukii* are critical for providing proteins for development of flies raised on fresh fruit, but that these microbial proteins are not essential for development of flies raised on a nutrient sufficient diet. 

In our study, we also observed that the effect of bacterial abundance produced at the level of individual taxa on larvae phenotype varied with diet. In a few cases, even the direction of correlation between the abundance of microbial taxa and the tested phenotype was opposite on different diets. In addition, using PCA, we observed that correlational effects microbial abundance (as an example at the family level) produced on measured metabolic (Appendix A) and fitness phenotypes (development rate and survival; Appendix A) varied between the diets. The correlation coefficients for the influence of all microbial taxa on measured metabolic phenotypes clustered together for the PR diet but not for other diets (Appendix A). The correlation coefficients between bacteria abundances and fitness phenotypes clustered for all but PA diets (Appendix A).

## 5. Conclusions

In conclusion, our work has demonstrated that the bacterial community within a larva is influenced by the parental bacterial community, bacterial community found in the diet, the diet itself, and the genotype of the host. Further, those bacterial communities correlated with host metabolic phenotypes both at the level of specific taxa and as a function of diversity of the community.

## Figures and Tables

**Figure 1 microorganisms-08-01972-f001:**
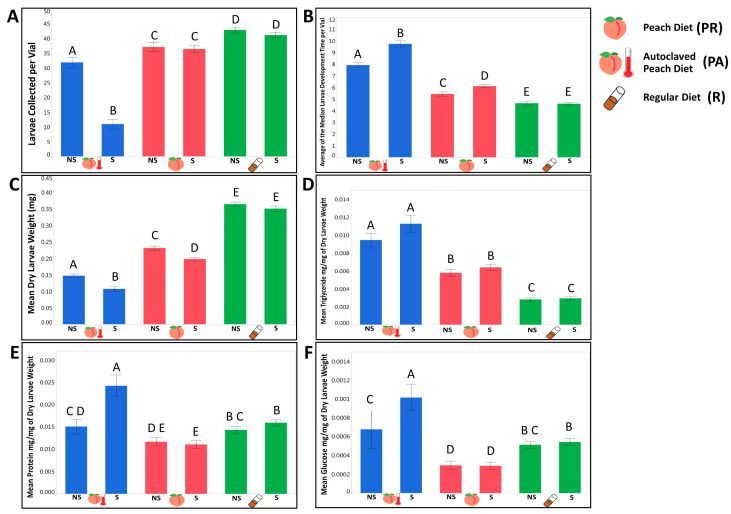
The presence of dietary and/or maternally inherited bacteria substantially impacts metabolic and life history phenotypes in flies. (**A**) Mean survival until the late 3rd instar stage is increased on lab diet and with the presence of bacteria on peach diets; (**B**) mean development time is decreased on a lab diet and with the presence of dietary and/or maternal bacteria on peach diets; (**C**) mean larval weight is higher on a lab diet and in the presence of bacteria on peach diets; (**D**) triglyceride concentrations by weight are reduced on a lab diet and in the presence of environmental bacteria on peach diets; (**E**) protein by weight is lower on a peach diet, in the presence of bacteria; and (**F**) glucose by weight is lowest on a peach diet in the presence of bacteria. Error bars indicated one standard error of the mean. S stands for sterilized larvae and NS stands for non-sterilized larvae. Bars that do not have common letters above them represent significantly different groups (*p* < 0.05) following a pairwise post hoc Student’s *t*-test.

**Figure 2 microorganisms-08-01972-f002:**
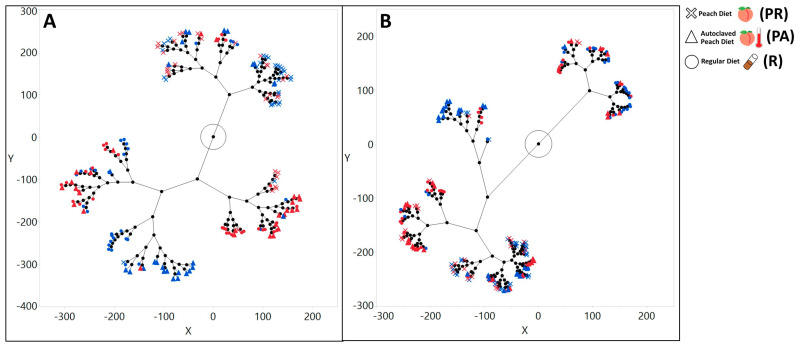
Larvae raised on peach and regular lab diets form distinct bacterial communities. Constellation plot based on hierarchal clustering of larvae bacterial community beta diversity distances. (**A**) Bray–Curtis distances indicate that symbiotic bacteria of the larvae raised on a peach diet is different from larvae raised on a regular diet. Symbiotic bacterial communities of PA raised larvae are more similar with R raised larvae than with PR raised larvae, especially if larvae were not sterilized. (**B**) Weighted Unifrac distances indicate that based on phylogenetic beta diversity distances PR and R diets form distinct communities. Samples from sterilized larvae are marked with a blue color and samples from non-sterilized larvae are marked with the red color. Samples raised on regular lab diet are marked with circled, on a peach diet with crosses and on peach autoclaved diet with triangles.

**Figure 3 microorganisms-08-01972-f003:**
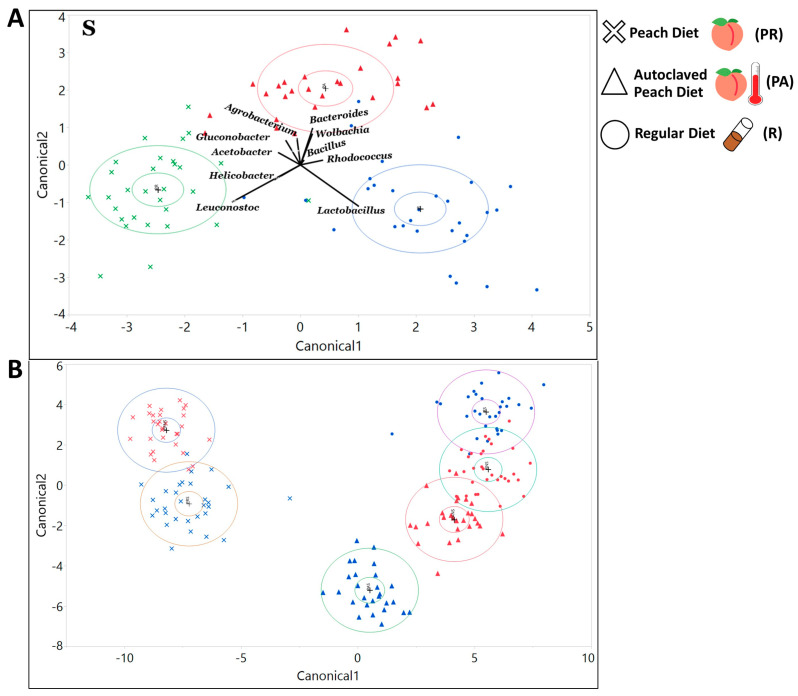
Microbial communities of the larvae raised on all of the diets could be differentiated by the abundance of dominant bacteria taxa. Discriminant analysis of symbiotic bacterial community based on taxa relative abundances in larvae. (**A**) The 10 most abundant bacterial genera found in sterilized larvae. Diet serves as a good predictor for bacterial communities differentiation. The length of the vector is correlated with the strength of the impact that it produced for the samples to be separated, in the vector direction, on the canonical plot. (**B**) The 100 most abundant ZOTU found in larvae. All diets and treatments (non-sterilized red and sterilized blue) form distinct communities, with the PR raised larvae being the most different from larvae raised on other diets.

**Figure 4 microorganisms-08-01972-f004:**
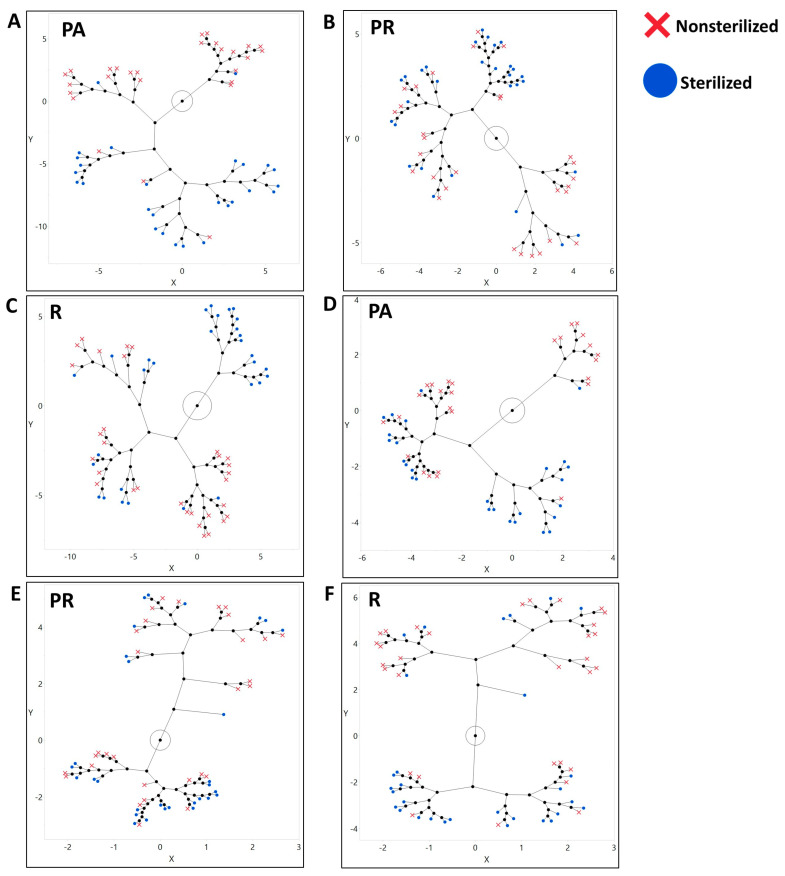
Inheriting parental bacteria influences the formation of larvae symbiotic bacterial community with the effect being unequal among the diets. (**A**–**C**) Bray–Curtis distances between NS and S larvae raised on PA, PR, and R diets, respectively. NS and S communities are clearly separated on (**A**) PA and (**C**) R diets, and (**B**) PR food is less distinct. (**D**–**F**) Weighted Unifrac distances. Larvae raised on (**A**) PA and (**C**) R diets form more distinct communities than larvae raised on (**B**) PR diet.

**Figure 5 microorganisms-08-01972-f005:**
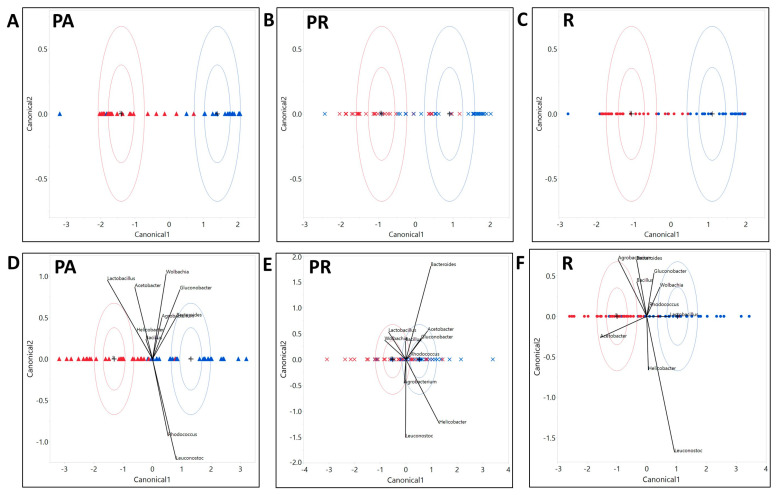
Microbial communities of the larvae raised with or without parental microbiota could be differentiated by the abundance of all bacteria taxa and by the abundance of the dominant taxa only. Discriminant analysis of symbiotic microbiota community based on taxa relative abundances in larvae from non-sterilized (red) and sterilized (blue) treatments. The analysis includes (**A**) All identified ZOTUs on a PA diet (**B**) on a PR diet, and (**C**) on an R diet. On all diets, treatment serves as a good predictor for a differentiation of the symbiotic bacterial communities. (**D**) The 10 dominant genera on a PA diet. (**E**) The 10 dominant genera on a PR diet. (**F**) The 10 dominant genera on an R diet. Treatment serves as a good predictor for the separation of bacterial communities on (**D**) PA and (**F**) R diets and is associated with the dominant bacteria taxa.

**Figure 6 microorganisms-08-01972-f006:**
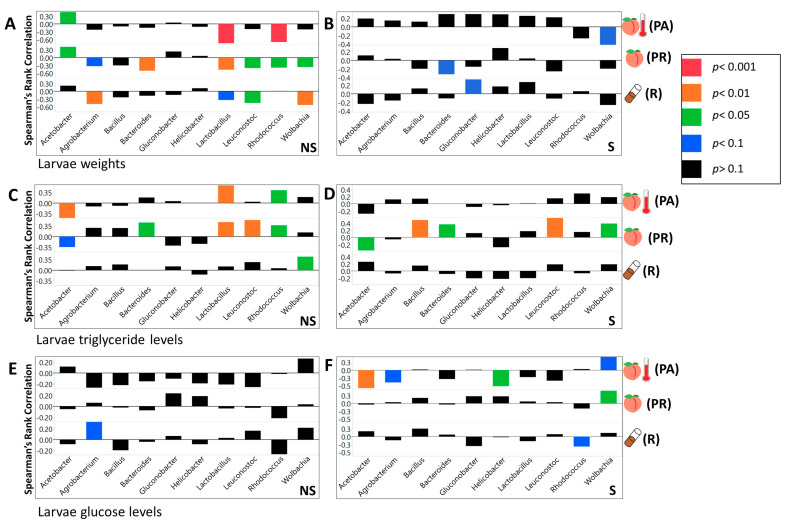
The correlations between the abundance of symbiotic bacterial taxa and larvae metabolic phenotypes vary with the diets. Spearman’s rank correlation coefficients between the abundance of 10 dominant bacterial genera and larvae phenotypes, on each diet. The color of the bars corresponds to the level of significance for each correlation (uncorrected for multiple comparisons). (**A**) Weight of NS larvae. (**B**) Weight of S larvae. (**C**) Triglyceride levels of NS larvae. (**D**) Triglyceride levels of S larvae. (**E**) Glucose levels of NS larvae. (**F**) Glucose levels of S larvae. The effect that symbiotic bacteria produce on all metabolic phenotypes may change with the diet.

**Figure 7 microorganisms-08-01972-f007:**
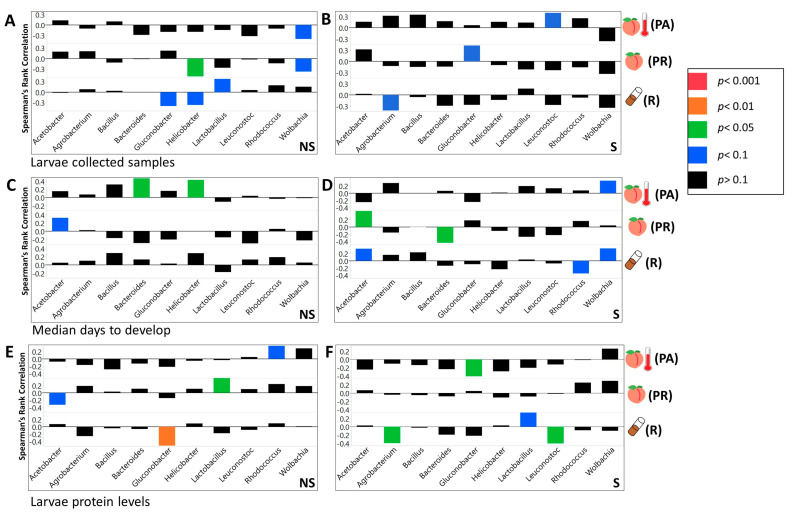
The correlations between the abundance of bacterial taxa and larvae life history traits and metabolic phenotypes vary with the larval diet. Spearman’s rank correlation coefficients between the abundance of 10 dominant symbiotic bacterial genera and larvae phenotypes on each diet. The color of the bars corresponds to the level of significance (uncorrected for multiple comparisons) for each correlation. (**A**) Total proportion of NS larvae reaching the 3rd instar stage; (**B**) total proportion of S larvae reaching the 3rd instar stage; (C) median number of days to reach pre-pupation stage for NS larvae; (**D**) median number of days to reach prepupation stage for S larvae; (**E**) protein concentrations of NS larvae; and (**F**) protein concentrations of S larvae. Symbiotic bacteria may influence life history traits and protein concentrations of the larvae differently depending on a diet.

**Table 1 microorganisms-08-01972-t001:** The difference in larvae phenotypes based on the diet. Comparison of larvae life history traits and metabolic phenotypes between larvae, from all 10 genetic lines, raised on regular lab diet (R), peach diet (PR), and autoclaved peach diet (PA). NS stands for non-sterilized larvae, S stands for sterilized larvae. Asterisks indicate the uncorrected significance of comparisons *p* < 0.001 *** and *p* < 0.01 **.

Treatment	Survival	Development	Weight	Triglyceride	Protein	Glucose
NS	R > PR **	R < PR ***	R > PR ***	R < PR ***	R > PR **	R > PR ***
S	R > PR **	R < PR ***	R > PR ***	R < PR ***	R > PR ***	R > PR ***
NS	PR > PA ***	PR < PA ***	PR > PA ***	PR < PA ***	PR < PA	PR < PA **
S	PR > PA ***	PR < PA ***	PR > PA ***	PR < PA ***	PR < PA ***	PR < PA ***

**Table 2 microorganisms-08-01972-t002:** The influence of parental microbiota on larvae phenotypes. Comparison of larvae life history traits and metabolic phenotypes between larvae, from all 10 genetic lines, raised on regular lab diet (R), peach diet (PR), and autoclaved peach diet (PA). NS stands for non-sterilized larvae, S stands for sterilized larvae. Asterisks indicate the uncorrected significance of comparisons *p* < 0.001 *** and *p* < 0.01 **.

Genetic Line	Diet	Survival	Development	Weight	Triglyceride	Protein	Glucose
All	PA	S < NS ***	S > NS ***	S < NS ***	S > NS	S > NS **	S > NS **
All	PR	S < NS	S > NS ***	S < NS ***	S > NS	S < NS	S > NS
All	R	S < NS	S > NS	S < NS	S > NS	S > NS	S > NS

**Table 3 microorganisms-08-01972-t003:** The contribution of diet, genotype, treatment, and their interactive effects on formation of larvae life history traits and metabolic phenotypes. VE stands for variance explained, by each independent variable. Asterisks indicate the significance of comparisons *p* < 0.001 ***, *p* < 0.01 **, and *p* < 0.05 *.

Independent Variable	Survival	Development	Weight	Triglyceride	Protein	Glucose
Diet	VE = 28.4% ***	VE = 47.5% ***	VE = 31.4% ***	VE = 41.4% ***	VE = 4.24% ***	VE = 17.8% ***
Genetic line	VE = 5.13% ***	VE = 4.98% ***	VE = 7.44% ***	VE = 2.36% **	VE = 5.71% ***	VE = 3.15%
Treatment	VE = 4.69% ***	VE = 2.41% ***	VE = 0.74% ***	VE = 0.36%	VE = 0.40% *	VE = 2.61% ***
Diet*Genetic line	VE = 3.13% ***	VE = 0.96%	VE = 2.88% ***	VE = 5.03% ***	VE = 5.09% ***	VE = 5.52%
Diet*Treatment	VE = 8.02% ***	VE = 1.41% ***	VE = 0.10%	VE = 0.07%	VE = 0.45%	VE = 3.55% ***
Genetic line*Treatment	VE = 0.98% *	VE = 2.82% ***	VE = 1.06% ***	VE = 1.81% *	VE = 3.53% ***	VE = 0.72%
Diet*Treatment*Genetic line	VE = 2.37% ***	VE = 0.82%	VE = 2.75% ***	VE = 4.81% ***	VE = 4.92% ***	VE = 2.37%

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
