# Peer review of "Influence of Lab Adapted Natural Diet and Microbiota on Life History and Metabolic Phenotype of Drosophila melanogaster"

_microorganisms, 2020, doi:10.3390/microorganisms8121972_

Round 1
Reviewer 1 Report
The present study investigates the effects of diet on microbiota, life-history traits and metabolic phenotype in Drosophila melanogaster larvae. In my opinion, the study is generally well performed and well analyzed. I have several comments that are listed below.
Abstract L9: I do not think that this paper is about obesity – I would suggest starting the abstract with a different sentence.
Introduction: L30-38: I suggest removing this paragraph.
Introduction: The authors should emphasize that D. melanogaster is a saprophytic insect that feeds on microbes growing on rotting fruits (unlike D. suzukii).
Materials and Methods:
L121-122: The authors should justify their selection of this particular diet (peaches) and procedure (why six days?). Moreover, please provide information on the weather conditions (at least the mean temperature) during the six days and the location.
L164: What was the volume of vials?
L127: The precise composition of the lab food should be stated here.
L142: I know that you refer to your previous publications but it would be better if you provide information on temperature, humidity and light/dark cycle in this manuscript as well.
L162-201: There is no information on rearing/developmental temperature, humidity, L:D regime etc.
L172-201: Phenotypes, such as developmental rate, weight, triglyceride and protein content etc. are affected by gender (e.g. Testa et al., 2013). Since the authors did not take this important factor into account (although it is relatively easy to determine larval sex), they should at least acknowledge it.
L182, L186, L188, L203: How many samples per specific group were used?
Results
L298-327: Since the authors assessed the triglyceride, protein and glucose concentrations in their experimental diets, I would suggest starting this part by comparing the macronutrient content of different diets (+ figure ??).
How do you explain differences in the macronutrient content between the autoclaved and non-autoclaved diet?
L305-308: I am not sure whether I understand this sentence correctly, but based on the Sup. Table 1, the autoclaved diet differs from the non-autoclaved diet in the macronutrient content. So what exactly do you mean when you claim then that the differences in the weight are caused by microorganisms in the food substrate? (This question is related to the previous one.)
L501: “Lactobacillales”.
L542-556: I would suggest rewriting this part – this is just a list of the numbers of correlations and interactions – perhaps present this information in a table.
Table 1, 2: Are the p-values corrected for multiple comparisons?
Figure 1: Please explain what the letters (in histograms) indicate. What statistical test was used to compare the data? Are the p-values corrected for multiple comparisons? Each figure legend should contain a description of the statistical test used.
Figs. 2, 4, 5. Individual symbols (crosses, triangles etc.) are too small.
Fig. 6 & 7: Are the p-values corrected for multiple comparisons? In addition, I would recommend highlighting only correlations with p-values (corrected for multiple comparisons) that are lower than the conventionally accepted threshold 0.05.
Fig. 6 & 7 (and elsewhere): Although, authors found several significant correlations between some bacterial taxa and phenotypes, however, this does not prove causation. I would recommend removing statements claiming that there is a direct influence of bacterial taxa on a given phenotype unless it was proved experimentally.
Reviewer 2 Report
After a long time, I have read an excellent piece of work entitled "Influence of lab adapted natural diet and microbiota on life history and metabolic phenotype of Drosophila melanogaster". It is not only well written but well described in detail. I have spent three hours on this paper, and it was worth reading. Profoundly appreciate authors for their hard work.
I would appreciate if the authors would shorten the introduction part. Remove unnecessary text in the introduction and add it in the discussion part.
Highly recommend its publication.
Author Response
Thank you for your recommendation.
Consistent with suggestions, we removed the paragraph L. 31-39 from the introduction.
Round 2
Reviewer 1 Report
The authors have satisfactorily addressed most of my comments.